# Analysis of Natural Heat Dissipation Capacity of Hydraulic Tank and Relevant Influencing Factors

Fugang Zhai [1,2,*], Xiaonan Wang [1], Zhiqiang He [1], Yu Chen [3], Zi Ye [1] and Jing Yao [1]

1   School of Mechanical Engineering, Yanshan University, Qinhuangdao 066004, China
2   Hebei Innovation Center for Equipment Lightweight Design and Manufacturing, Qinhuangdao 066004, China
3   School of Mechanical Engineering, Nanjing Institute of Technology, Nanjing 211167, China
*   Correspondence: zhaifugang@ysu.edu.cn; Tel.: +86-136-13387160

**Abstract:** This paper aims to study the natural heat dissipation capacity of a hydraulic tank during its miniaturization revolution. A theoretical model of heat dissipation was built up on the basis of experimental analysis. Then, the natural heat dissipation power was deduced and shown to be relevant. Influencing factors were analyzed, which were the oil height proportion, design proportion, volume, material type, and wall thickness. The results showed that the heat dissipation power is proportional to the height of the oil in the tank. The power increases with the height proportional coefficient k2, while it first decreases and then increases with the length proportional coefficient k1. The lengthwise coefficient obviously has a more significant effect. The influence degree of reduction methods on natural heat dissipation is in the following order: length reduction > equal proportion reduction > height reduction > width reduction. Additionally, when the thermal conductivity λ is greater than 10 W/(m·K), the material and wall thickness of the tank slightly influence the heat dissipation capacity; otherwise, the influence is evident.

**Keywords:** hydraulic tank; miniaturization; natural heat dissipation; influence factors

## 1. Introduction

Hydraulic systems are widely used in aerospace [1,2], robotics [3,4], engineering machinery [5,6], and other fields [7–9]. With the rapid development of science and technology, all fields have set a higher request for the power-to-weight ratio. The miniaturization and light weight of hydraulic components and systems can not only reduce the weight of the equipment and improve its endurance and mobility, but also help to achieve energy conservation and emission reductions [10,11]. A hydraulic tank accounts for a large proportion of the volume and weight in the hydraulic system. It is one of the components with the light weight and greatest miniaturization potential in hydraulic systems [12]. Because of the energy loss of hydraulic system, the temperature of hydraulic oil will rise. The main sources of energy losses are pumps [13], motors [14–16], and valves. The reduction of the volume of the hydraulic tank may lead to the deterioration of the natural heat dissipation capacity of the tank, which will then affect the thermal equilibrium temperature of the hydraulic system and bring hidden dangers to the good functioning of the hydraulic system [17,18]. Therefore, with the aim to provide guidelines for the miniaturization and weight reduction of the hydraulic tank, this paper studies the natural heat dissipation capacity of the hydraulic tank and its influencing factors.

At present, domestic and foreign scholars' research on the natural heat dissipation of hydraulic tanks mainly focuses on the optimization of heat dissipation capacity. For example, Zhang at al. [19] redesigned the tank from the aspects of the tank capacity, structure, and technique and improved the natural heat dissipation capacity of the hydraulic tank. Guo at al. [20] designed a new type of constant-temperature tank based on semiconductor refrigeration technology, which can control the temperature at 37 ± 1 °C. Lv at

al. [21] applied fins to the outer wall of the tank, designed four new structural tanks, used Fluent to simulate the heat dissipation process of the tank, and studied its heat dissipation efficiency. By simulating the heat transfer performance of the hydraulic tank, Liang at al. [22] proposed to use phase change silicone material to wrap the side of the tank to improve the heat transfer performance of the hydraulic tank. Guan at al. [23] designed a tank with a replaceable heat exchange plate by taking advantage of the high latent heat and high heat storage density of phase change materials, which optimized the heat dissipation capacity of the hydraulic tank. Wu at al. [24] studied the thermal balance of the hydraulic steering system of a tractor, and they pointed out that increasing the volume of the tank will also result in the increase of the total oil volume and heat dissipation area of the system, thereby reducing the system temperature rise rate and thermal equilibrium temperature. In addition, Zhang at al. [25] considered the convective heat transfer between the oil and the oil tank, the heat conduction between the inner and outer walls of the oil tank, and the convective heat transfer between the outer wall of the oil tank and the air when calculating the heat dissipation capacity of the hydraulic tank, and they improved the thermal model of the hydraulic tank. Lan at al. [26] analyzed the thermal model of an aircraft fuel tank, then further introduced the gas in the upper part of the tank and abstracted the wall of the tank as the "Upper wall", "Lower wall", and several "Side walls". They derived the heat balance equations between the walls and a more complete dynamic thermal model of the tank.

Aiming at the development demand for the miniaturization of hydraulic tanks, this paper deduces the calculation formula of the natural heat dissipation power by an analysis via mathematical modeling and studies the natural head dissipation capacity of the hydraulic tank and its influencing factors in the process of miniaturization, so as to provide theoretical guidelines for the miniaturization of hydraulics tank and the weight reduction of the hydraulic system.

## 2. Analysis of Heat Dissipation in a Hydraulic Tank

There are various energy losses in the hydraulic system, most of which are absorbed by the hydraulic oil in the form of heat energy, causing the temperature of the hydraulic oil to rise [27]. As an important part of the natural heat dissipation of the hydraulic system, the hydraulic tank has a large volume and a long oil flow path, resulting in an unbalanced oil temperature distribution at different positions in the oil tank. To analyze the severity of this unbalanced phenomenon, a thermal balance experiment was carried out with a hydraulic tank of an injection molding machine. The installation position of the thermocouple is shown in Figure 1. Thermocouple 1 is installed near the outlet, and Thermocouple 2 is installed near Inlet 2.

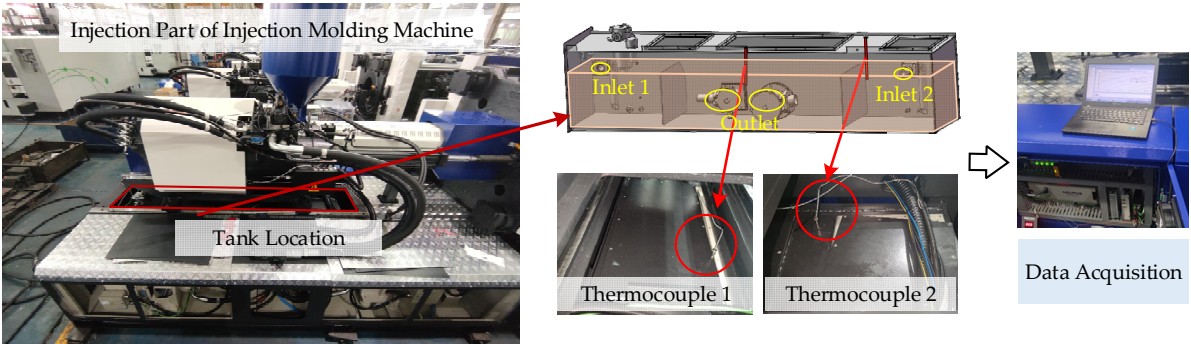

**Figure 1.** Heat balance experiment of injection molding machine tank.

The distance between Thermocouples 1 and 2 was 0.826 m, and the test time of the tank was 4.5 h. By fitting and predicting the test curve, the temperature change curve of the inlet and outlet when the tank reaches the thermal balance state was obtained, as shown in Figure 2. The temperature of the inlet and outlet of the tank tended to stabilize after the

machine ran for 10 h. The thermal balance temperature was about 50 °C. The temperature difference of the inlet and outlet was 0.4 °C.

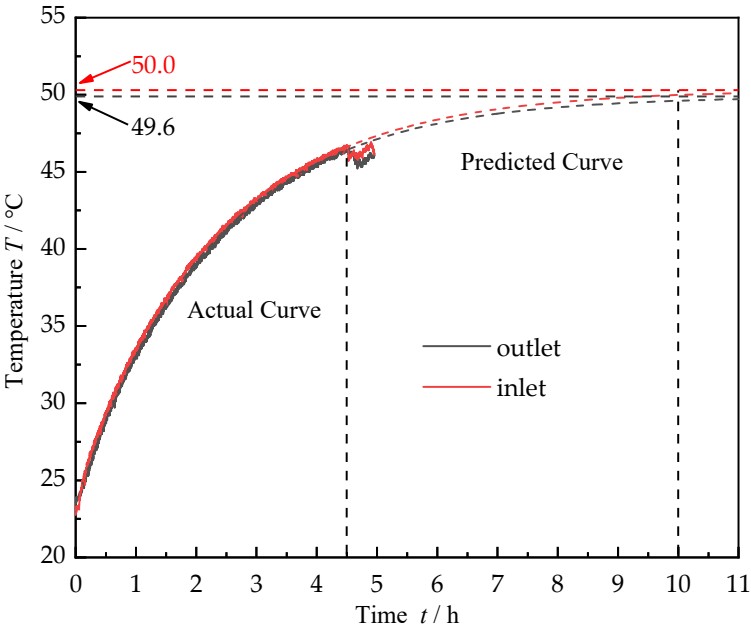

**Figure 2.** Curve of temperature change with time in tank.

The temperature difference between the inlet and outlet was small, so each part of the tank was simplified into different nodes during the theoretical deduction, and the nodes were connected by heat conductors. The parameters of the nodes and heat conductors represent the temperature of the medium and the heat transfer mode between the media, respectively. At the same time, considering that a certain amount of air is usually reserved at the top of the hydraulic tank, therefore, the mathematical modeling of the hydraulic tank was divided into the "Oil contact part" and the "Air contact part" according to the different contact media of its inner wall, as shown in Figure 3.

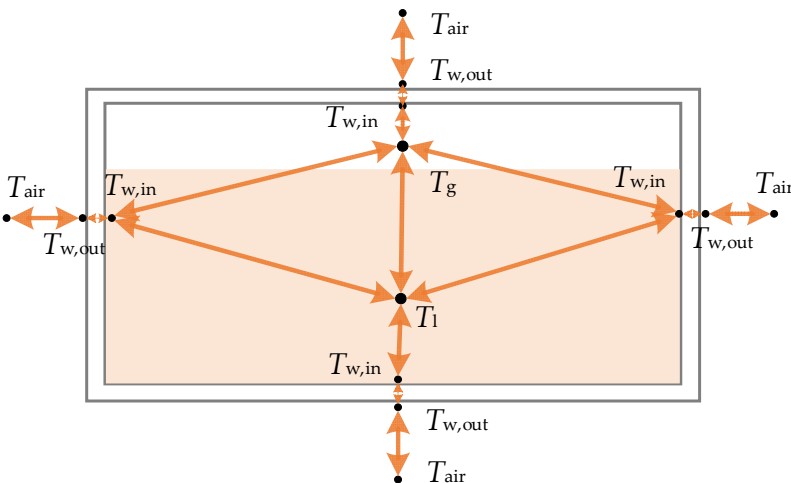

**Figure 3.** Natural heat dissipation model of hydraulic tank.

Based on the above assumptions, the natural heat dissipation process of the tank is: convective heat transfer between oil and inner wall, heat conductive between the inner wall and the outer wall, convective heat transfer between the outer wall and the air, convective heat transfer between the oil and air in the tank, and convective heat transfer between the air and the inner wall in the tank. In Figure 3, $T_{w,in}$ represents the inner wall temperature

of hydraulic tank, $T_{\text{w,out}}$ represents the outer wall temperature of the tank, $T_{\text{g}}$ represents the gas temperature in the hydraulic tank, $T_1$ represents the oil temperature, and $T_{\text{air}}$ is the gas temperature outside the hydraulic tank.

## 3. Deduction of Mathematical Model for Natural Heat Dissipation in a Hydraulic Tank

The natural heat dissipation process of the hydraulic tank is complicated, and the following assumptions were made for the convenience of mathematical modeling analysis:

1. The temperature of the hydraulic oil, gas, and each wall is a lumped parameter.
2. The radiation heat transfer of the hydraulic tank was not considered.
3. We only considered the horizontal attitude of the hydraulic tank, regardless of its special conditions.

### 3.1. Heat Dissipation on the Oil–Tank Wall Contact Area

The contact area between the oil and the inner wall of the tank is:

$$A_{\text{oilw}} = 2(a+b)h + ab \tag{1}$$

where $a$ is the length of the hydraulic tank (m); $b$ is the width of the hydraulic tank (m); $h$ is the height of the oil in the hydraulic tank (m).

The power of the convective heat transfer between the oil and the tank inner wall is:

$$P_1 = K_1(T_l - T_{\text{w,in}})A_{\text{oilw}} \tag{2}$$

where $K_1$ is the heat transfer coefficient between the oil and the inner wall (W/(m$^2$·K)).

The power of heat conduction in the tank wall is:

$$P_2 = \lambda \frac{T_{\text{w,in}} - T_{\text{w,out}}}{\delta} A_{\text{oilw}} \tag{3}$$

where $\lambda$ is the thermal conductivity of the material of the hydraulic tank (W/(m·K)); $\delta$ is the wall thickness of the hydraulic tank (m).

The power of convective heat transfer between the tank outer wall and the ambient air is:

$$P_3 = K_2(T_{\text{w,out}} - T_{\text{air}})A_{\text{oilw}} \tag{4}$$

where $K_2$ is the convection heat transfer coefficient between the air and the outer wall (W/(m$^2$·K)).

The natural heat dissipation in the hydraulic tank can be approximated as a steady conduction process. The flux of heat through each medium is identical. Therefore, $P_1 = P_2 = P_3$, then the power of the heat dissipation of the "Oil contact part" can be deduced:

$$P_{\text{oil}} = \frac{(T_1 - T_{\text{air}})[2(a+b)h + ab]}{\frac{1}{K_1} + \frac{\delta}{\lambda} + \frac{1}{K_2}} \tag{5}$$

### 3.2. Heat Dissipation on the Oil–Air Contact Area

The contact area between the oil and air in the tank is:

$$A_{\text{l,g}} = ab \tag{6}$$

The contact area between the air in the tank and the inner wall is:

$$A_{\text{airw}} = 2(a+b)(H-h) + ab \tag{7}$$

where $H$ is the height of the hydraulic tank (m).

The power of convection heat transfer between the oil and the ambient air is:

$$P_4 = K_3 (T_1 - T_g) A_{\text{airw}} \tag{8}$$

where $K_3$ is the convective heat transfer coefficient of the air and oil in the tank (W/(m²·K)).

The power of the convective heat transfer between the tank inner wall and the ambient air is:

$$P_5 = K_4 (T_g - T_{w,\text{in}}) A_{\text{airw}} \tag{9}$$

where $K_4$ is the convective heat transfer coefficient between the air and the inner wall of the tank (W/(m²·K)).

The power of the heat conduction of the tank inner wall and outer wall is:

$$P_6 = \lambda \frac{T_{w,\text{in}} - T_{w,\text{out}}}{\delta} A_{\text{airw}} \tag{10}$$

The power of the convective heat transfer between the tank outer wall and the ambient air is:

$$P_7 = K_2 (T_{w,\text{out}} - T_{\text{air}}) A_{\text{airw}} \tag{11}$$

The analysis of the "Oil contact part" and "Air contact part" is identical, so the power of the heat dissipation of the "Air contact part" is:

$$P_{\text{air}} = \frac{(T_1 - T_{\text{air}})[2(a+b)(H-h) + ab]}{\frac{2(a+b)(H-h)+ab}{K_3 ab} + \frac{1}{K_4} + \frac{\delta}{\lambda} + \frac{1}{K_2}} \tag{12}$$

### 3.3. Natural Heat Dissipation Capacity of Hydraulic Tank

The total capacity of natural heat dissipation in a hydraulic tank is composed of the heat dissipation power in the "Oil contact part" and in the "Air contact part", namely the sum of Equations (5) and (12):

$$P = P_{\text{oil}} + P_{\text{air}} = \frac{(T_1 - T_{\text{air}})[2(a+b)h + ab]}{\frac{1}{K_1} + \frac{\delta}{\lambda} + \frac{1}{K_2}} + \frac{(T_1 - T_{\text{air}})[2(a+b)(H-h) + ab]}{\frac{2(a+b)(H-h)+ab}{K_3 ab} + \frac{1}{K_4} + \frac{\delta}{\lambda} + \frac{1}{K_2}} \tag{13}$$

## 4. Discussion

From Equation (13), the heat dissipation capacity closely depends on the oil height proportion, the geometrical design proportion, the volume, the material type, and the wall thickness of the hydraulic tank. These factors were analyzed in detail based on the data in Table 1.

**Table 1.** Calculation parameters.

| Parameter | Value | Parameter | Value |
|-----------|-------|-----------|-------|
| $T_1$ | 65 °C | $K_1$ | 500 W/(m²·K) |
| $T_{\text{air}}$ | 20 °C | $K_2$ | 12 W/(m²·K) |
| $V$ | 500 L | $K_3$ | 7 W/(m²·K) |
| $\delta$ | 6 mm | $K_4$ | 7 W/(m²·K) |

### 4.1. Influence of the Oil Height

When the hydraulic system is working, the oil height in the tank is constantly changing due to the asymmetry of the hydraulic cylinder with the rod cavity and the rodless cavity. The influence of the oil height ratio ($k_3$) on the natural heat dissipation of the tank was analyzed by changing the oil height, and the universality was verified by changing the design proportion of the tank, as shown in Figure 4. The design proportion in this paper refers to the ratio of the length, height, and width of a rectangular tank, which ranges from 1:1:1 to 3:2:1.

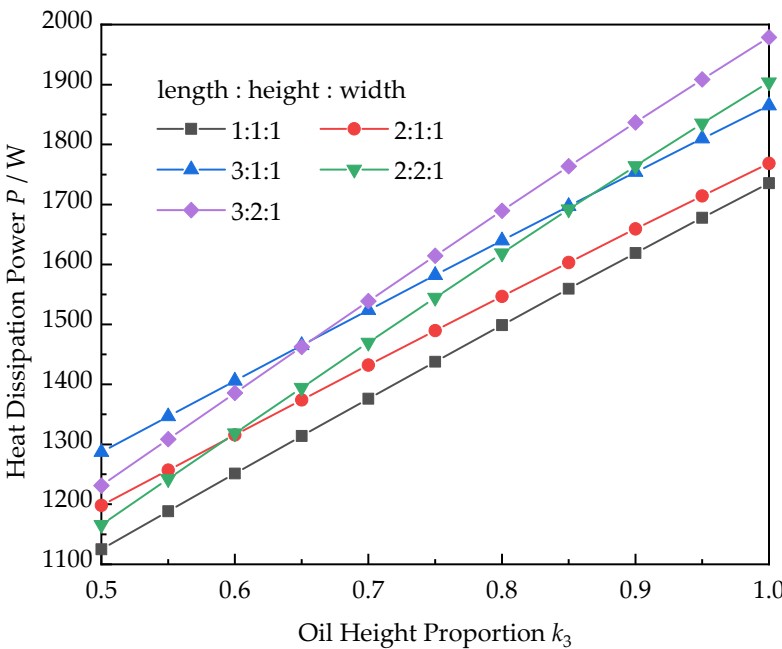

**Figure 4.** Heat dissipation power with oil height under different length-height-width ratios.

It can be seen from Figure 4 that the natural heat dissipation power of the tank is proportional to the liquid level height. That is, the higher the liquid level is, the greater the heat dissipation power is. This also shows that the capacity of the heat conduction of the oil is greater than that of the air. In addition, the value of the natural heat dissipation power is different when the design proportion of the tank is different. Their relationships will be further elucidated in the next section.

### 4.2. Influence of the Geometrical Design Proportion

The width ($b$) of the tank was selected as the dimensional benchmark. Then, other dimensions can be expressed as: $a = k_1 b$, $H = k_2 b$, $h = k_3 H = k_2 k_3 H = 0.8\, k_2 b$ ($k_3$ is usually 0.8), where $k_1$ is the length proportion coefficient of the hydraulic tank and $k_2$ is the height proportion coefficient of the hydraulic tank. Then, Formula (13) becomes:

$$P = \frac{(T_1 - T_{\text{air}}) \times (1.6 k_1 k_2 + 1.6 k_2 + k_1) b^2}{\frac{1}{K_1} + \frac{\delta}{\lambda} + \frac{1}{K_2}} + \frac{(T_1 - T_{\text{air}}) \times (0.4 k_1 k_2 + 0.4 k_2 + k_1) b^2}{\frac{0.4 k_1 k_2 + 0.4 k_2 + k_1}{K_3 k_1} + \frac{1}{K_4} + \frac{\delta}{\lambda} + \frac{1}{K_2}} \quad (14)$$

The relationship between the natural heat dissipation power of the hydraulic tank and the design proportion of the tank is shown in Figure 5. It shows that, in the range of the design ratios of the length, height, and width from 1:1:1 to 3:2:1, when the design ratio is 3:2:1, the maximum heat dissipation power is 1689.5 W. When the design ratio is 1.1:1:1, the minimum heat dissipation power is 1496.4 W. In addition, if the length proportional coefficient $k_1$ is constant, the power increases with the increase of the height ratio coefficient $k_2$. The maximum power increases by 3.0% compared with the minimum power. If $k_2$ is constant, the power decreases first and then increases with the increase of $k_1$. The maximum power increases by 9.4% compared with the minimum power. Therefore, the length proportional coefficient $k_1$ of the hydraulic tank has a more obvious influence on the natural heat dissipation capacity of the tank.

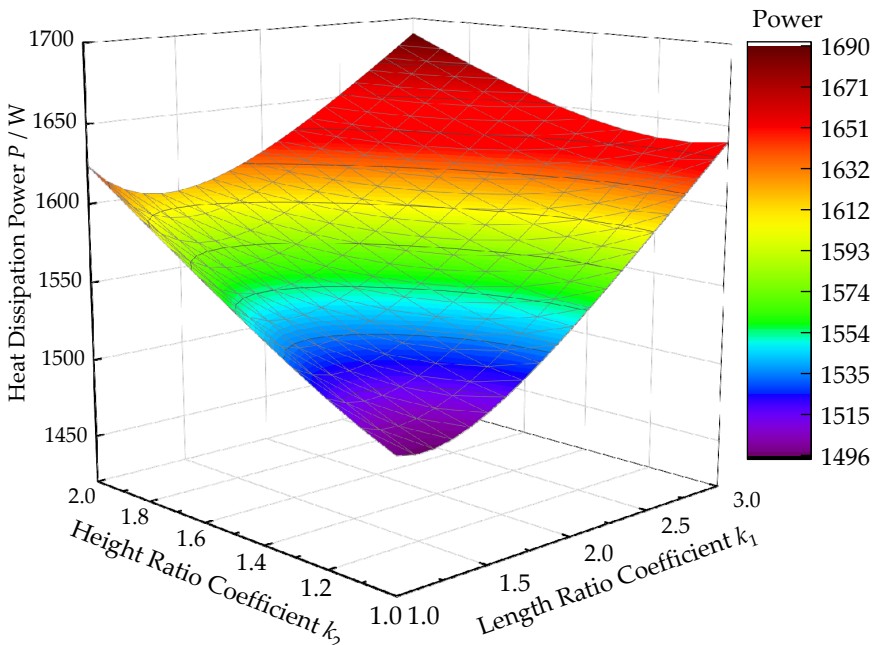

**Figure 5.** Curve of heat dissipation power with design proportion.

The length of the tank affects the flow path of the oil in the tank. The longer the flow path is, the more conducive to the realization of degassing and impurity removal function [28]. The comparison of the tank parameters for the design ratios of 3:1:1 and 3:2:1 is shown in Table 2. It shows that the tank with a heat dissipation power of 3:2:1 is only 3.0% higher than that of the tank with a design ratio of 3:1:1, but the length of the tank is reduced by 20.7%. Therefore, considering the flow path of the oil in the tank, the hydraulic tank with the design ratio of 3:1:1 is better. Therefore, it was used in the calculation and analysis below.

**Table 2.** Comparison of tank parameters under different design ratios.

| Design Ratios | 3:1:1 | 3:2:1 | Change |
|:---:|:---:|:---:|:---:|
| $P/W$ | 1639.9 | 1689.5 | +3.0% |
| $a/mm$ | 1651 | 1310 | $-20.7\%$ |

*4.3. Influence of Volume Reduction Method of Hydraulic Tank*

The miniaturization of the hydraulic tank has different reduction methods. They are the length reduction, width reduction, height reduction, and proportional reduction. The width is the smaller of a and b. In reality, the convection coefficient will not only change with the geometric changes, but also with the external environment, such as the temperature, wind speed, etc. However, this article assumes them to be constant for simplicity. The volume of the hydraulic tank was reduced by N-times by different reduction methods, and the natural heat dissipation power of the hydraulic tank was obtained as shown in Figure 6.

It can be seen that different reduction methods have different effects on the power. The descending order is length reduction > equal proportion reduction > height reduction > width reduction. The differences between the length reduction and width reduction in Figure 6 are based on that the starting dimensional ratio of 3:1:1. Therefore, in the process of the miniaturization of the hydraulic tank, the size in the width direction should be reduced first to ensure the natural heat dissipation capacity of the tank. In addition, if the model started with a 1:1:1 ratio, there should not be any difference when reducing the volume by a or b in the generic case.

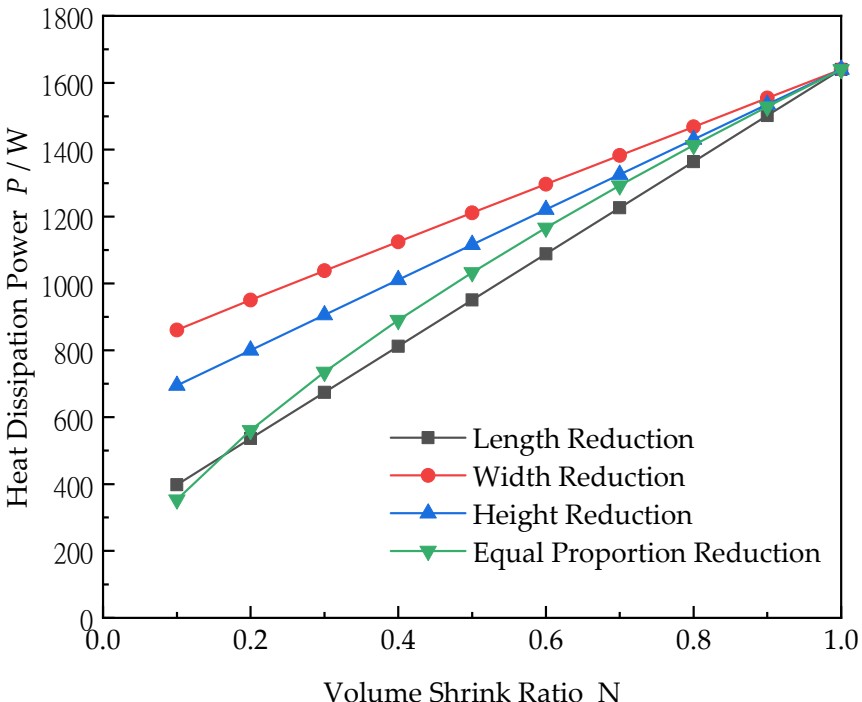

**Figure 6.** Influence of different reduction methods on heat dissipation power.

### 4.4. Influence of Material and Wall Thickness of Hydraulic Tank

The traditional hydraulic tank is generally made of metal materials with a large volume and high weight. Under the development trend of weight reduction, non-metallic hydraulic tanks have gradually attracted attention. Different materials have different thermal conductivities, which thereby affects the natural heat dissipation capacity of the hydraulic tank. The influence of the thermal conductivity on the heat dissipation capacity of the tank is shown in Figure 7.

It can be seen from Figure 7 that, when the thermal conductivity $\lambda$ is less than 1 W/(m·K), the heat dissipation power increases exponentially. When $\lambda$ is 1~10 W/(m·K), the power increases slowly. When $\lambda$ is greater than 10 W/(m·K), the power tends to be flat. Thus, if $\lambda$ is greater than 10 W/(m·K), material substitution has little effect on the heat dissipation capacity of the tank. If $\lambda$ is less than 10 W/(m·K), the material with a greater thermal conductivity should be selected to improve the heat dissipation capacity of the tank.

In addition to the influence of the material (that is, the thermal conductivity $\lambda$), the wall thickness $\delta$ of the tank also affects the heat dissipation power. With regard to the setup of the hydraulic tank material shown in Table 3, among them, Q235 and aluminum alloy (5052) are common metal materials for hydraulic tanks, and cross-linked polyethylene and nylon are the common non-metal materials. Furthermore, the two materials with a thermal conductivity of 1 W/(m·K) and 10 W/(m·K) are user-defined materials.

**Table 3.** Hydraulic tank material and its thermal conductivity.

| Materials Properties | Materials Name | Thermal Conductivity (W/(m·K)) |
| --- | --- | --- |
| Metal | Q235<br>Aluminum Alloy (5052) [29] | 50<br>235 |
| Non-Metal | Cross-Linked Polyethylene [30]<br>Nylon [31] | 0.4<br>0.7 |
| User-Defined | $\lambda = 1$<br>$\lambda = 10$ | 1<br>10 |

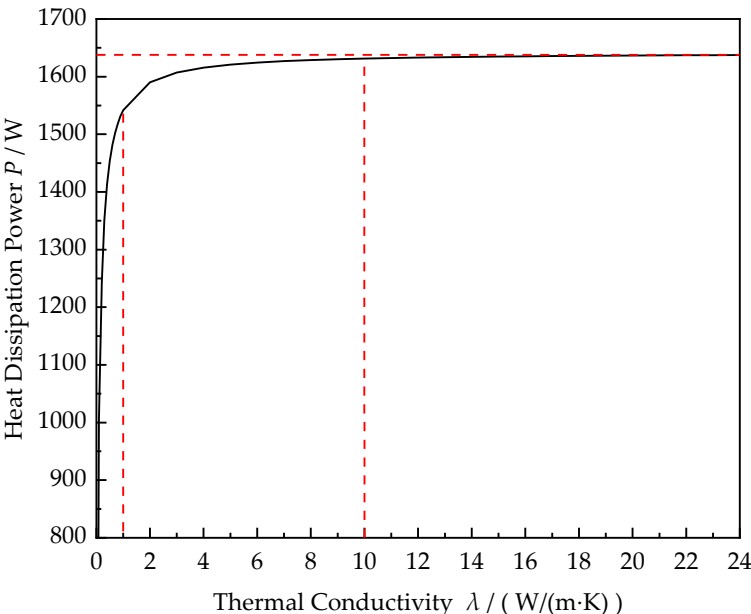

**Figure 7.** Curve of heat dissipation power of tank with thermal conductivity.

The relationship between the heat dissipation power and wall thickness of the above six materials is shown in Figure 8. It can be observed from Figure 8 that the power of the hydraulic tank decreases with the increase of the wall thickness with the same material. However, the influence of the tank wall thickness on the power varies greatly with different materials. For example, when the material is Q235, aluminum alloy, or $\lambda = 10$, the wall thickness of the tank has little effect on the heat dissipation power; when the material is cross-linked polyethylene, nylon, or $\lambda = 1$, the wall thickness of the tank has a great impact on the power.

According to Figures 7 and 8, when $\lambda$ is greater than 10 W/(m·K), the material type and wall thickness have little effect on the heat dissipation capacity of the hydraulic tank. When $\lambda$ is less than 10 W/(m·K), the material type and wall thickness have a great influence on the heat dissipation capacity of the hydraulic tank.

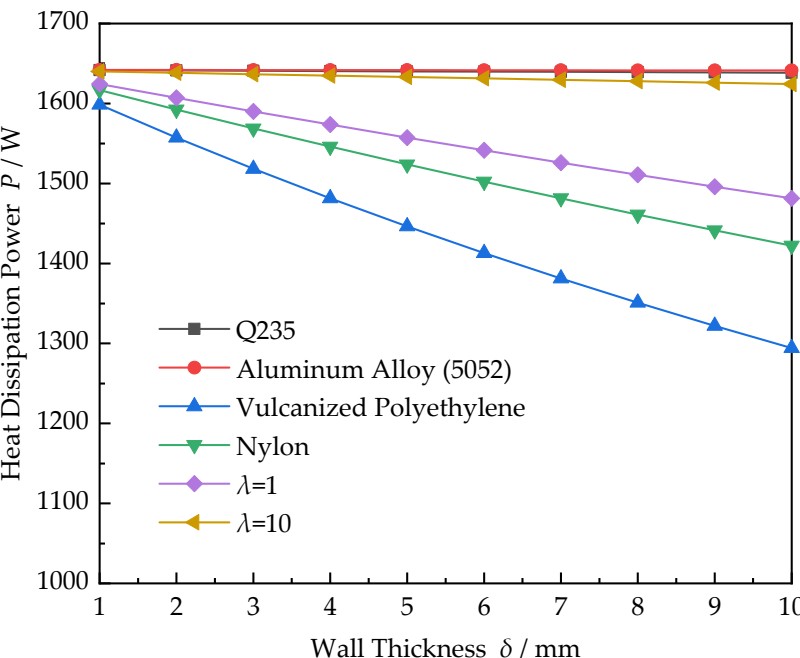

**Figure 8.** Curve of heat dissipation power of tank with wall thickness.

## 5. Conclusions

In this paper, a mathematical model of hydraulic tank dissipation was established to deduce the natural heat dissipation. On this basis, the influence of the oil height ratio, tank design ratio, tank volume reduction mode, material, and wall thickness on the natural heat dissipation capacity was analyzed. The results showed:

1.  The heat dissipation power of the hydraulic tank is proportional to the oil height. That is, the higher the oil height ratio is, the greater the natural heat dissipation power is. This means the capacity of the heat conduction capacity of oil is greater than that of the air.
2.  When the length ratio coefficient $k_1$ is constant, the natural heat dissipation power of the hydraulic tank increases with the height ratio coefficient $k_2$. When $k_2$ is constant, the power decreases first and then increases with the $k_1$. Compared with $k_2$, the length ratio coefficient $k_1$ of the hydraulic tank has a greater impact on the natural heat dissipation capacity.
3.  The order of the effect of the reduction method on the natural heat dissipation power of the hydraulic oil tank is length reduction > equal proportion reduction > height reduction > width reduction.
4.  Moreover, when the thermal conductivity $\lambda$ of the material is greater than 10 W/(m·K), the material and wall thickness of hydraulic tank have little effect on the heat dissipation capacity of the hydraulic tank. When $\lambda$ is less than 10 W/(m·K), the effect is great. Therefore, for metal tanks, replacing materials with those with better thermal conductivity cannot effectively improve the heat dissipation capacity of the tank.

**Author Contributions:** Conceptualization, F.Z. and Y.C.; methodology, F.Z. and J.Y.; software, X.W.; validation, Z.Y. and Y.C.; formal analysis, Z.H.; investigation, Z.Y.; data curation, Z.H.; writing—original draft preparation, X.W.; writing—review and editing, X.W.; project administration, F.Z. and J.Y. All authors have read and agreed to the published version of the manuscript.

**Funding:** This research was funded by the National Key Research and Development Program of China (2018YFB2000703) and the Postdoctoral Research Foundation of Hebei Province (B2022003022).

**Institutional Review Board Statement:** Not applicable.

**Informed Consent Statement:** Not applicable.

**Data Availability Statement:** Not applicable.

**Conflicts of Interest:** The authors declare no conflict of interest.

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
