# Peer review of "Analysis of Natural Heat Dissipation Capacity of Hydraulic Tank and Relevant Influencing Factors"

_machines, doi:10.3390/machines10110991_

Round 1

Reviewer 1 Report

Overall, the article is well written.

The results of research and analyzes are of practical importance - they may be useful for designers of hydraulic systems.

The article may be published after completing the literature review on the sources of energy losses in the hydraulic system. Therefore, it should be written that the main sources of energy losses in the hydraulic system are pumps, motors and valves. Please refer briefly to the literature:  https://doi.org/10.3390/ en14185669 , https:// doi.org/10.3390/en15010298 , https:// doi.org/10.17559/TV-20161119215558 , https:// doi.org/10.1016/j.procs.2021.08.195 and ect.

Nevertheless, the author / authors should in the introduction write more about the sources of energy losses in the hydraulic system (indicate pumps, hydraulic motors and valves).

Reviewer 2 Report

The paper presents a relatively simple model for heat transfer from a hydraulic reservoir and then explores some of the effects of parameters. The simple model is OK, as it means that understanding all the parameters is a tractable task, but unfortunately this is not done in this paper. The authors picked some base parameters and then varied some and plotted the results. This means that the results given are applicable for their particular parameters but can not be widely generalized. I would have expected a more rigorous selection of meaningful nondimensional parameters and used the equation 13 given to explore their significance. This would expand what is currently of niche interest (i.e. only those with a problem with the same base parameters) to a very useful paper with wide applicability to all reservoirs. The paper is acceptable, but I was a bit disappointed because it would only take a little bit more to make it an excellent paper.

This paper is relatively well written. There are a few typos and some idiomatic language that are not standard, but the point of the writing is clear. I'd recommend one more proof-read. A few things are out of order. I found myself confused and rereading parts to work it out, only to find the missing information on a later page.

Some specific comments:
Sect 3 defines the model. The equations used a reasonable but assumptions and limitations should be stated.
Sect 4.1: redefines K3 as oil height ratio. Should this be k3? These ratios should be defined (I see they're defined in Sec 4.2, but should be defined here) and I'd recommend using a different symbol rather than lower case k, to avoid confusion with the upper case K.
Figure 4: refine in caption or legend what the different colors mean. What does 1:1:1 mean?
Section 4.1 says that heat dissipation is different when the proportions change, but doesn't specify how. E.g. the heat dissipation is greatest with greater width and length than height. I see that you explain this in Sec 4.2, so maybe say that this will be explained in the next section.
Sec 4.2: h=k3H=0.8k2b. Should this be k3k2b? I don't understand why you later say "k3 is usually 0.8". Isn't this a variable? You can assume an arbitrary constant value, but make this clear.
Why is k2 limited to 2.0 but k1 goes to 3?
In Section 4.3, you should clarify that the convective coefficients will change with geometrical changes in reality, but you are assuming them constant for simplicity
Sect 4.2 "the length of the tank affects the flow path". You should clarify that the effect of this on heat transfer is not included in your model.
Sect 4.3 Since your model (eq13) doesn't have any difference between length and width (a and b), there should not be any difference if you reduce the volume by a or b in the generic case. You should clarify that the differences in Fig 6 are based on the starting dimensional ratios (i.e. the ratios with N=1). If you started with a 1:1:1 ratio, the "length" and "width" lines should overlap.  Also, please specify which ratio you started with. You should also clarify that according to your model, you should first minimize the width as defined by the smaller of a and b, not the direction perpendicular to the flow path.
Sect 4.4 You should clarify that the 1 W/(mK) value for conductivity is only for your particular base parameters. Is there a non-dimensional number that can be used to relate the knee in the conductivity curve to the other parameters to make a more general curve? Presumably, form Eq 13 this occurs when delta/lambda << 1/K1+1/K2 and << 2(a+b)(H-h)+ab/K3ab+1/K4+1/K2. If the oil part dominates, then (lambda/delta*(1/K1+1/K2) might be a good choice. If K2<<K1, this simplifies to lambda/(delta*K2)
Table 3:  I'm not sure what the "User defined" rows are meant to show.
